# Optimizing Sensor Placement for Event Detection: A Case Study in Gaseous Chemical Detection

**DOI:** 10.3390/s25082397

**Published:** 2025-04-10

**Authors:** Priscile Fogou Suawa, Christian Herglotz

**Affiliations:** Department of Computer Engineering, Brandenburg University of Technology Cottbus-Senftenberg, 03046 Cottbus, Germany; christian.herglotz@b-tu.de

**Keywords:** event detection, gaseous substances, industrial monitoring applications, influence of sensor location, multi-objective optimization, sensors, sensors fusion, supervised learning

## Abstract

In dynamic industrial environments, strategic sensor placement is key to accurately monitoring equipment and detecting critical events. Despite progress in Industry 4.0 and the Internet of Things, research on optimal sensor placement remains limited. This study addresses this gap by analyzing how sensor placement impacts event detection, using chemical detection as a case study with an open dataset. Detecting gases is challenging due to their dispersion. Effective algorithms and well-planned sensor locations are required for reliable results. Using deep convolutional neural networks (DCNNs) and decision tree (DT) methods, we implemented and tested detection models on a public dataset of chemical substances collected at five locations. In addition, we also implemented a multi-objective optimization approach based on the non-dominated sorting genetic algorithm II (NSGA-II) to identify optimal sensor configurations that balance high detection accuracy with cost efficiency in sensor deployment. Using the refined sensor placement, the DCNN model achieved 100% accuracy using only 30% of the available sensors.

## 1. Introduction

Event detection is crucial in various fields, including smart production, mobility, homes, cities, and Industry 4.0 [1,2]. In industrial applications, detected events can encompass sounds, temperature fluctuations, speed levels, lighting conditions, the presence of gases, and other critical indicators. Sensors are essential in these applications, serving as primary tools for collecting event-related data [3]. Since events can occur over varying durations and intensities, the location and number of sensors in an event-prone area are crucial to obtaining data representations that adequately capture these events.

This paper examines how positioning and the number of sensors affect the efficiency of detecting gaseous substances. We use an open-access dataset presented by Vergara et al. [4], which consists of measurements from an array of gas sensors operating in a controlled wind tunnel environment. The dataset was designed to simulate real-world conditions by exposing metal oxide (MOX) gas sensors to controlled mixtures of different gases at various concentrations. These MOX sensors, mainly from the Figaro series, detect gases based on changes in electrical resistance caused by interactions with gas molecules. Although collected in 2013, this dataset remains a valuable reference for the gas detection community due to its controlled experimental conditions and widespread use for evaluating machine learning-based classification models [5,6]. Furthermore, advancements in MOX sensor technology [7], such as compact multi-sensor modules (e.g., MiCS-6814 and Sensirion SGP41), provide new opportunities for applying our optimization approach to modern sensor platforms. These developments highlight the ongoing relevance of our methodology in optimizing sensor deployment for gas detection.

Using this dataset, we applied supervised learning techniques, including deep convolutional neural networks (DCNNs) and decision trees (DTs), to train models and evaluate detection accuracy at different locations. Our experimental results indicate considerable variations in detection accuracies based on sensor locations: locations with higher gas concentrations yielded superior results, achieving up to 80% and 97% accuracy for the DT and DCNN models, respectively. When data from multiple sensors were merged, detection accuracy improved to 100% at all locations tested, regardless of the detection algorithm used.

While these results underscore the importance of sensor location for reliable gas detection, one major challenge in practical applications is reducing the number of sensors while maintaining high detection accuracy. In this study, we propose to optimize the sensor placement through a multi-objective optimization approach using the non-dominated sorting genetic algorithm II (NSGA-II) algorithm [8,9]. The optimization process’s results enabled the configuration of the model trained on fused data to be reduced from eight to a maximum of three sensors with an accuracy of between 99% and 100% at the various locations. These results were achieved by using the most appropriate sensors at each location. This optimization balances high detection precision with the cost of sensor deployment by recommending minimal sensor placements at optimal locations. Our findings contribute to the growing need for efficient, scalable sensor deployment strategies to meet the detection accuracy requirements without incurring excessive costs. This approach is a foundation for advancing event detection applications in resource-limited or cost-sensitive environments, with implications for broader industrial monitoring applications.

This paper is organized as follows: Section 2 provides the background and literature review. Section 3 discusses the context and dataset. Section 4 presents the methodology, while Section 5 covers the results and discussion. Finally, Section 6 concludes the paper.

## 2. Background and Literature Review

Detecting events with sensor networks is crucial in various fields, including environmental monitoring, industrial safety, and managing critical infrastructure. These systems can capture real-time information, monitor ambient conditions, and detect anomalies or specific events, such as the presence of pollutants or hazardous gases or changes in temperature or humidity [10]. In chemical detection, sensors are used to identify and quantify the presence of gaseous compounds that may represent a health or environmental hazard. Gas sensor systems, such as those based on metal oxides, are commonly deployed to monitor the presence of volatile chemical compounds in closed or open environments [11]. These sensors can detect a wide range of gases, with varying sensitivities depending on the material used. However, detection accuracy is highly dependent on several factors, including the type of sensor, its sensitivity, and its position in the space being monitored [12].

The authors in [13] propose a data fusion approach for the rapid detection and location of faults in industrial plants using wireless sensor networks. They present two Bayesian strategies, a three-layer architecture, and a two-layer architecture, designed to exploit sensor trustworthiness metrics to improve performance. Simulations of an active subsea oil and gas production system indicate that the two-layer architecture results in the lowest mean square error, highlighting its superior efficiency. In [14], the authors investigate an innovative state estimation technique within an advanced distributed framework designed to reduce computational complexity while detecting multiple sensor faults in hydrogen-blended natural gas pipelines. Based on the ensemble Kalman filter (EnKF) and called partial-distributed multi-sensor fault detection, isolation, and accommodation, this novel method has demonstrated effectiveness in highly nonlinear, high-dimensional systems experiencing multiple simultaneous sensor faults.

The study [15] describes the development of an electronic nose consisting of eight custom-made sensors using pure poly(3-hexylthiophene) (P3HT) and various doped materials. The goal is to electronically detect gases exposed to these sensors and assess the accuracy of gas classification. Resistance variations for each sensor were measured over time and processed using three identification techniques: principal component analysis (PCA), linear discriminant analysis (LDA), and k-nearest neighbor analysis (kNN). The results show that LDA is the most reliable and efficient method, achieving 100% prediction accuracy, compared to 93.52% for PCA and 73.14% for kNN. The authors [16] presented the development of a Long-Range (LoRa [17])-based sensor network for Air Quality Monitoring (AQM) and gas leakage detection. Combining a commercial gas sensor with a resistance measurement channel for graphene chemo resistive sensors allows the system to calculate an Air Quality Index based on the concentration of reducing species such as volatile organic compounds (VOCs) and CO, while also detecting $NO_{2}$, a significant air pollutant. The graphene sensor tested with the LoRa nodes can detect $NO_{2}$ pollution within 5 min and monitor sudden changes. Paper [18] introduced a novel model-based multi-sensor fault detection, isolation, and accommodation (MM-SFDIA) technique designed to address multiple simultaneous sensor faults in large-scale distributed systems. The approach uses a distributed filtering framework with multiple local ensemble Kalman filters (EnKFs). Simulated data generated from a numerical solution of the transient flow model for natural gas demonstrate the architecture’s effectiveness, showcasing high accuracy and low execution time in detecting and isolating multiple sensor faults.

The study [19] focuses on the applications of air-borne methods for inspecting natural gas pipelines, with the primary objective of testing an unmanned aerial vehicle (UAV) equipped with a remote-sensing methane detector for natural gas leak detection across the pipeline network. Measurement data were collected and analyzed using machine learning techniques, which facilitated the identification of spatially correlated regions with elevated methane concentrations. However, field experiment results indicated that identifying areas with increased methane concentrations proved to be significantly challenging, although they remained detectable. The experiments suggest that UAV flights should be conducted at lower altitudes. Overall, the findings demonstrate that UAV monitoring can serve as an effective tool for the preliminary identification of potentially faulty sections in gas pipelines.

In this context, where sensor networks play a key role in event detection, several studies [20,21,22] have highlighted the importance of optimal sensor location to maximize detection coverage and accuracy. A judicious distribution of sensors allows to improve detection efficiency while reducing the costs associated with installing and operating these systems. Optimizing the location of sensors has, therefore, been the subject of a great deal of research using a variety of approaches.

The authors in [23] focus on developing a methodology to select and position hazardous gas detectors within complex industrial layouts. The methodology utilizes preliminary computational fluid dynamics (CFD) simulations to assess the extent of hazardous gas clouds. An optimization approach is established using a twofold algorithm. The first algorithm identifies optimal sensor positions to maximize detection frequency for a specified number of sensors. The second algorithm employs multicriteria decision analysis based on the reference point method to determine the optimal economic number of sensors, facilitating the most cost-effective strategy.

In [24], the authors discuss the importance of gas detection systems as a critical layer of protection in process safety, highlighting the significance of leak scenario probability and detector reliability in optimizing gas detector placement. A stochastic programming (SP) optimization method is proposed using the particle swarm optimization (PSO) algorithm. A case study conducted on a diesel hydrogenation refining unit demonstrates the effectiveness of this method in improving detection efficiency.

The authors in [25] highlight that continuous or regularly scheduled monitoring has the potential to rapidly identify environmental changes. The physical placement of sensors, along with the chosen technology and operating conditions, significantly affects the performance of the monitoring strategy. Chama, an open-source Python (2.7, 3.4, 3.5, or 3.6) package integrating mixed-integer and stochastic programming formulations to optimize sensor locations, is utilized here to design sensor networks for monitoring airborne pollutants and assessing water quality in distribution systems.

Yuan Zi et al. [26] introduce a mathematical method to determine the optimal placement of sensors to establish a cost-effective monitoring strategy for the earliest possible alarm in response to micro-seismic events. This method focuses on minimizing detection time and enhancing overall monitoring performance.

Several optimization approaches are commonly used to solve multi-objective problems similar to sensor placement optimization in the literature. Approaches such as evolutionary algorithms, particle swarm optimization (PSO) [27], and stochastic search methods such as simulated annealing [28] have shown interesting performance in complex environments. However, each of these methods has specific strengths and limitations depending on the context of the problem. The NSGA-II algorithm was chosen for this study because it is well suited for multi-objective problems requiring a compromise between conflicting objectives [29,30]. In our case, minimizing the number of sensors is crucial while maximizing detection accuracy. NSGA-II can generate a set of optimal solutions representing different trade-offs on the Pareto front, offering greater flexibility in the selection of final solutions.

Although significant advancements have been made in chemical gas detection, notable gaps still exist concerning sensor placement optimization. Most studies focus on isolated aspects of detection without addressing the crucial issue of optimal sensor location in complex environments. Furthermore, few works integrate multi-objective optimization methods, which are essential for balancing detection accuracy and sensor deployment costs. Utilizing experimental data from metal oxide sensors in a wind tunnel, this study aims to address existing gaps by proposing an approach that optimizes sensor configuration, determining specific locations for effective chemical gas monitoring.

## 3. Context and Dataset

### 3.1. Context

Nowadays, the monitoring of industrial machinery or equipment incorporates intelligent components. In particular, smart sensors for collecting real-time data will be used to understand the behavior of the different machines [31]. In this paper, we analyze a scenario that involves several sensors located at different positions to evaluate the impact that the position of the sensors has on the quality of the monitoring of equipment. Figure 1 shows the scenario where there are *m* different sensors and *n* different places to assess the sensors. The experiment is carried out with the sensors at each position. After positioning the sensors and starting the experiment, the sensor node receives the data and transfers it to a central server, studying the machine’s behavior. The monitoring module in this scenario includes different learning models to detect or predict events because one model might be more accurate in one position than another.

The case studied in this work to evaluate the influence of the position of the sensors according to the above scenario is that of monitoring a wind tunnel by a process of the detection of gaseous chemical substances. The dataset created by Alexander Vergara et al. [4], consisting of measurements from various chemical sensors placed at different position, was used for the experiments. Section 3.2 provides an overview of this dataset.

### 3.2. Dataset

The dataset employed in this work is presented in [32] and was generated by Alexander Vergara et al. [4]. It contains time series data from a chemical sensing platform exposed to different gas conditions in a turbulent wind tunnel. The measurement platform consists of nine portable chemosensory modules, each equipped with an identical set of 8 metal oxide-based gas sensors [32], which makes the 72 metal oxide gas sensors used to record data at the different locations. At each location, ten chemical gases are released into the wind tunnel. These gases, **acetone, acetaldehyde, ammonia, butanol, ethylene, methane, methanol, carbon monoxide, benzene, and toluene,** represent the labels of the data and, therefore, will be the outputs of the detection models. The dataset contains sensor measurements evaluated at five different operating temperatures and three different wind speeds to induce different turbulence levels in the wind tunnel. However, in the context of this paper, which aims to highlight the influence of sensor position, the experiments are conducted on only part of the data for this very large dataset. In particular, those collected from the sensors at the first five locations have the following settings: the temperature attained by setting the constant voltage of 4.0 V to the sensors’ heating element and wind flow speeds of 0.10 m/s induced by the rotational speeds of the tunnel exhaust fan of 1500 rpm (25 Hz).

Each measurement contains 75 time series, where the first three attributes represent information on time, temperature, and humidity, and the last 72 correspond to the 72 sensors, recorded continuously over 260 s at a sampling rate of 100 Hz (samples per second), resulting in 26,000 samples per sensor for a single setup. Given that the dataset includes measurements from multiple setups or experimental runs, the total number of samples across all setups exceeds this value by far. Each sensor provides 12-bit readings over all the samples. The dataset is categorized based on the ten gases that must be identified. The authors in [4] show the measurement setup and provide a more detailed analysis and discussion of recording time series and their graphical representations.

To evaluate the quality and consistency of the data, we examined two key aspects:Behavior Across All Sensors: Figure 2 shows data collected in the chemosensory module 2 at two different positions (P1 and P4) from eight sensors, each tracking a distinct signal. The general trends in both graphs are similar, suggesting that the gas exposure events at P1 and P4 follow a comparable sequence. The periodic increases and decreases in sensor values could indicate a reaction to varying gas concentrations. In P4, some sensors (e.g., Sensors 6 and 8) exhibit higher peak values than in P1. This suggests that gas concentrations at P4 might be slightly higher, or environmental factors (e.g., airflow) influence the readings. Looking at fluctuations in sensor values over time, the data at P4 have slightly more noise in some sensor signals, especially in the mid-range values (e.g., Sensor 5). P1 data, in contrast, looks smoother with fewer abrupt fluctuations.Consistency Among the different modules: The graphs in Figure 3 represent data from Sensor 2 recorded across nine chemosensory modules (Module 1 to Module 9) at two positions, P1 and P4. Both locations exhibit distinct step-like changes in sensor readings over time, indicating abrupt shifts in gas concentrations. P4 displays higher variability and noise in sensor signals than P1, particularly in the mid-range values. Most modules have similar trends, responding similarly to gas concentration changes. This suggests that the sensor is generally reliable across different modules. Some modules (e.g., Module 8 in P4) show higher fluctuations, which may indicate sensitivity differences or noise issues. P4 appears to have more noise than P1, possibly due to environmental factors (e.g., turbulence or gas diffusion patterns).

The dataset is well structured, with clear patterns and minimal deviations between sensor data of the same type collected from different modules, indicating high-quality data collection under controlled conditions. The variability observed across different locations underscores the importance of the environmental context, as sensors are sensitive to their surroundings. The way the sensor data change suddenly in steps instead of smooth transitions suggests that the measured gas concentration is shifting in distinct phases rather than gradually increasing or decreasing. These transitions could correspond to real-world events or conditions, like a sudden appearance, disappearance, or change in gas concentration levels. Such behavior is valuable for analysis because it provides clear markers or features that can be used to identify and characterize events.

## 4. Methodology

The problem addressed in this study can be formulated as a multi-objective optimization problem, where we have the following objectives. Objective 1: Minimize sensor deployment and maintenance costs, which in this study means reducing the number of active sensors $N_{\mathrm{act}}$ required for effective environmental monitoring. Objective 2: Maximize event detection, in this case by maximizing the overall accuracy $O_{\mathrm{acc}}$, measured as the average of the detection accuracies at each sensor position. To solve this problem, we use the evolutionary optimization technique, particularly the NSGA-II [8,9] algorithm, which combines these two contradictory objectives. This algorithm is particularly well suited to solving complex problems where optimal solutions lie on a Pareto front [33], offering several possible sensor configurations that optimize cost and accuracy simultaneously.

Figure 4 describes the sequential steps followed during this work using the two approaches presented in [34], which led to the results presented in this paper. In the next subsections, we will discuss each module in detail.

### 4.1. Pre-Processing and Data Fusion

As mentioned in the previous section, 75 time series of 26,000 points were recorded at each setup run, i.e., a matrix of size 26,000 × 75. The pre-processing step consisted of removing the repetitive data, particularly the temperature and humidity, which were constant for each example. The time column was also removed, as it does not represent information characterizing an event. Finally, each example consists of 72 attributes corresponding to the 72 sensors on the data measurement equipment.

Multi-sensor data fusion refers here to the simultaneous use of the data from all the sensors, i.e., the 72 attributes in the input vector of the learning models. It is, therefore, a concatenation fusion, the resulting sample matrix $X_{fused}$ being of size 26,000 × 72 (72 = 8 (the number of sensors) × 9 (the number of modules of the data acquisition system)). Single-sensor data fusion involves concatenating single-sensor data from all modules. In this case, the sample matrix size is 26,000 × 9 for the training models.

### 4.2. Principal Component Analysis

The pre-processing step is followed by calculating the components (PCA) [35], which will be used as input for the decision tree model. Applying PCA to the original dataset leads to principal component coefficients, also known as loadings, which reflect the contribution of each of the original coordinates to the new coordinates and to scores, which are representations of a data matrix $X_{fused}$ in the principal component space. For an *n* by *p* data matrix $X_{fused}$, where the rows *n* correspond to the observations and the columns *p* to the variables, the coefficient matrix $X_{coefs}$ is *p* by *p*, each column containing the coefficients of a principal component. The score matrix $X_{scores}$ is of size *n* by *p*, where the rows correspond to the observations and the columns to the components [35].

This method is utilized as a feature extraction technique to generate inputs for the decision tree model. The PCA method identifies and projects the most significant dimensions of the dataset, simplifying and reducing its size by analyzing the relationships among the observations and their associated dimensions [36].

### 4.3. Decision Tree

Many algorithms are proposed in the literature to build decision trees. In this experiment, a decision tree classifier, a supervised learning algorithm [37], is used. It is built from the Matlab Classification Learner, where the tree predicts classifications based on predictors or features for each input data. Predicting starts with the top node. The predictor values are checked for each decision to decide which branch to follow. When the branches reach a leaf node, the data are ranked. The following parameters have been set for training the models:Model type: Fine tree, which is a decision tree with many leaves that makes many fine distinctions between classes.Maximum number of splits: 500, which defines the maximum number of splits or branch points to control the depth of the tree.Splitting criterion: Gini diversity index, which specifies the measure of the splitting criterion for deciding when to split nodes. It determines the purity of a specific class after it has been split according to a particular attribute. The best split increases the purity of the sets resulting from the split [38].Validation: Fivefold cross-validation is used to obtain a better insight into the prediction accuracy for the new data. It divides the training data into five random parts. It forms 05 new trees and then examines the predictive accuracy of each new tree on the data not included in the formation of that tree. This method gives a good estimate of the predictive accuracy of the resulting tree since it tests the new trees on new data.

The output of the decision tree model is a classification result per time sample, where each sample is assigned 1 of 10 gas labels listed in Section 3.2. The model is trained to detect and classify these gases based on sensor data, either from individual sensors or fused multi-sensor data.

### 4.4. DCNN

The DCNN model implemented in this study was inspired by the model presented in [39], to which we made some modifications to fit our dataset. As modifications, we updated the size of the input vector and the size of the convolutional layer core to 1 × 10; we also reduced the depth of the network and added a dropout layer to prevent the network from overfitting. Figure 5 shows the final configuration of the networks implemented. Two different DCNNs were implemented to handle single-sensor and multi-sensor input cases. In the single-sensor model, the input layer consists of nine features (corresponding to data from the same sensor in the nine modules). In the multi-sensor model, the input layer consists of 72 features (of all eight sensors). Each model was trained independently using the same training methodology, convolutional layers, batch normalization, dropout, and max-pooling. The structure in the single-sensor model contains only two convolutional layers, while three were used for the multi-sensor model. The deeper structure in the multi-sensor model (including an additional convolutional layer) accounts for the increased number of input features, allowing for better feature extraction across sensors. The training options are as follows: minibatch size: 2000; max epochs: 50; optimizer: Adam; and learning rate: 0.001. The output classes represent the 10 gases to be detected, listed in Section 3.2.

### 4.5. NSGA-II Optimization

In formulating an optimization problem for sensing tasks, it is common to include a performance threshold [40] as a constraint to ensure the system achieves a certain quality (in this case, accuracy). Many practical detection problems require meeting a minimum threshold to ensure the safety or reliability of a system, such as a minimum detection accuracy. However, we chose an approach that does not define such thresholds to provide greater flexibility in sensor selection. Instead of imposing arbitrary thresholds that may not be suitable for all sensing environments, the optimization algorithm automatically determines the optimal configuration. This enables us to explore the solution space fully without constraining our results with predefined criteria. Consequently, we can find an optimal balance between the number of sensors and accuracy more flexibly. This approach adapts better to changing or uncertain conditions and does not rely on initial threshold calibration, which may not be applicable in all scenarios. As a result, it enhances model generalization, especially in cases where the detection requirements are varied or imprecise. The NSGA-II Algorithm 1 is a widely used multi-objective optimization method that refines populations using evolutionary principles while preserving a diverse set of solutions. It is structured around three main components [8,9]:Non-dominated sorting: NSGA-II sorts candidate solutions into several Pareto fronts. A Pareto front contains so-called ‘non-dominated’ solutions, meaning no other solution on this front simultaneously outperforms a given solution for all objectives. NSGA-II categorizes solutions into Pareto fronts based on dominance relationships. A solution $s_{1}$ dominates another solution $s_{2}$ if $\forall i\in \left\{1,\ldots ,M\right\},\;f_{i}\left(s_{1}\right)\leq f_{i}\left(s_{2}\right)\;\mathrm{and}\;\exists j\in \left\{1,\ldots ,M\right\}\;\mathrm{such}\;\mathrm{that}\;$$f_{j}\left(s_{1}\right)<f_{j}\left(s_{2}\right)$, where $f_{i}\left(s\right)$ represents the *i*th objective function. The first Pareto front contains non-dominated solutions (no solution is strictly better in all objectives). The second front contains solutions dominated only by those in the first front, and so on.Crowding distance: The computation of the crowding distance involves arranging the population of solutions based on the values of each objective function in ascending order. After this sorting, the boundary solutions with the smallest and largest function values are given an infinite distance value. To maintain solution diversity, NSGA-II computes the crowding distance metric, which estimates how close a solution is to its neighbors. The crowding distance $d_{i}$ of a solution $x_{i}$ is given by(1)$$d_{i}=\sum_{m=1}^{M}\left(\frac{f_{m}^{i+1}-f_{m}^{i-1}}{f_{m}^{\max}-f_{m}^{\min}}\right),$$
where $f_{m}^{i+1}$ and $f_{m}^{i-1}$ are the function values of adjacent solutions in the sorted list of objective *m*, and $f_{m}^{\max},f_{m}^{\min}$ are the maximum and minimum values in that front. Larger crowding distances ensure better distribution across the Pareto front. Solutions in sparse regions are favored during selection.Genetic operators: NSGA-II uses genetic algorithm operators, namely selection, crossover, and mutation, to evolve the population over generations. Selection uses a binary sorting process based on Pareto dominance and crowding distance. Simulation binary crossover (SBX) is generally used to generate offspring solutions, and polynomial mutation is used to introduce small variations and maintain genetic diversity.
**Algorithm 1** NSGA-II algorithm overview. 1:Initialize population $P_{0}$ of size *N* 2:Evaluate the objective functions for each individual in $P_{0}$ 3:Perform non-dominated sorting on $P_{0}$ to determine Pareto fronts 4:Compute crowding distances for individuals 5:**for** each generation t=1,2,… **do** 6:    Select parents using binary tournament selection 7:    Apply crossover and mutation to generate offspring $Q_{t}$ 8:    Evaluate the objective functions for each individual in $Q_{t}$ 9:    Merge populations: $R_{t}=P_{t}\cup Q_{t}$10:    Perform non-dominated sorting on $R_{t}$ to determine Pareto fronts11:    Compute crowding distances for individuals12:    Select the best *N* individuals for the next generation $P_{t+1}$13:**end for**14:Return the final Pareto-optimal set

Let *x* be a binary vector, $x\in {\left\{0,1\right\}}^{n\times m}$, where *n* is the number of sensor positions available, and *m* is the number of sensor types at each position. Each element of the vector $x_{ij}$ indicates whether sensor *j* at position *i* is active (value of 1) or inactive (value of 0). *k* is the accuracy (Equation (Equation 5)) of a detection model trained on data from a single type of sensor. The objective functions to be optimized here are represented as follows: Concerning the first objective, we minimize the number of active sensors to reduce deployment and maintenance costs:(2)$$N_{\mathrm{act}}\left(x\right)=\sum_{i=1}^{n}\sum_{j=1}^{m}x_{ij}$$The aim is to minimize this sum, which represents the total number of sensors activated in the network. This number inherently accounts for both deployment and maintenance costs, as each additional sensor contributes to the overall expenses of installation, operation, and upkeep. Deployment costs include purchasing, calibration, and installation, while maintenance costs cover servicing, recalibration, power consumption, and replacements. Since these costs scale with the number of sensors, minimizing the total number of sensors effectively minimizes all associated costs.Maximize the overall accuracy of gas detection to ensure effective environmental monitoring:(3)$$O_{\mathrm{acc}}\left(x\right)=-\frac{1}{n}\sum_{i=1}^{n}\left(\frac{\sum_{j=1}^{m}x_{ij}\cdot k_{ij}}{\sum_{j=1}^{m}x_{ij}}\right)$$
where $k_{ij}$ is the accuracy of sensor *j* at position *i*. The overall accuracy is the average of the detection accuracies at each position, weighted by the sensor activity at each position. The negative sign in front of the function represents the need to maximize accuracy, as multi-objective optimization algorithms seek to minimize objective functions.Constraint: at least one sensor must be active in each critical monitoring zone to guarantee minimum detection throughout the monitored zone:(4)$$\sum_{j=1}^{m}x_{ij}\geq 1,\;\forall i=1,\ldots ,n$$We use the NSGA-II algorithm to solve this multi-objective optimization problem. This algorithm finds a set of optimal solutions (called the Pareto front), where each solution represents a compromise between the number of sensors activated and the overall detection accuracy. From these solutions, it is possible to select the configuration that maximizes accuracy while minimizing the number of sensors.

The algorithm is designed to investigate various combinations of sensors, considering their locations within the wind tunnel and their corresponding accuracy at each position. The overall accuracy is calculated based on all the active sensors at any given time. The algorithm operates as follows:Evaluation Function: at each iteration, the solutions generated by NSGA-II (composed of a certain number of active or inactive sensors at each position) are evaluated according to Equations (Equation 2) and (Equation 3).Optimization process [9]:
–Initialization: An initial population of solutions is randomly generated. Each solution represents a set of sensors activated or deactivated at different positions.–Pareto sorting: The solutions are ranked according to their performance on the two objectives. Non-dominated solutions are identified and ranked on the first Pareto front.–Crossover and mutation: At each iteration, the solutions are combined (crossover) and modified (mutation) to generate new sensor configurations while respecting the defined objectives.–Selection of optimal solutions: At the end of each generation, solutions are selected based on Pareto sorting and crowding distance, ensuring a good compromise between objectives and a diversity of proposed solutions.The algorithm is run over several generations until convergence is reached, resulting in optimal solutions representing trade-offs between the number of sensors and detection accuracy. These solutions recommend sensor combinations that optimize the detection system’s cost and performance.The Matlab 2023a implementation of the NSGA-II algorithm was used with a population size of 50, 100 generations, a crossover rate of 0.9, and a mutation rate of 0.05.

## 5. Results and Discussion

In this section, we first present the performance of the implemented detection algorithms in Section 5.1, demonstrating how sensor position affects the detection of various gases. We then present the results of the NSGA-II Multi-Objective Optimization in Section 5.2. Finally, we present the performance of DCNN using the optimal sensor sets resulting from NSGA-II in Section 5.3.

### 5.1. Performance of the DCNN and Decision Tree Detection Algorithms

Different models were trained using the data recorded at each of the five positions being evaluated. The dataset was randomly divided as follows: 90% for training and 10% for testing. The models were evaluated using the accuracy metric (*k*) defined in Equation (Equation 5) where TP, TN, FP, and FN stand for True Positive, True Negative, False Positive, and False Negative, respectively [41]:(5)$$k=\frac{TP+TN}{TP+TN+FP+FN}$$

This section presents the different results obtained from each method used. The models were trained using on fused data from all sensors and the data from each type of sensor individually, using the dataset presented in Section 3.2. Figure 6 and Figure 7 show the accuracy (Equation (Equation 5)) of the models trained using the DCNN and DT methods, respectively. The X-axis names the sensors, and the Y-axis shows the accuracy.

Decision tree models trained on PCA-generated features show varying levels of accuracy for the different sensors, with a significant drop compared to using the fused features of all sensors. The accuracies hover around 40–80% for individual sensors, and combining all sensors results in a notable increase in accuracy, reaching close to 100%. The reliance on manually crafted features significantly loses discriminatory information when using one sensor, leading to lower individual performance. This indicates that feature engineering may struggle to retain the full expressive power of the raw data.

The DCNN model demonstrates much higher accuracy across individual sensors, ranging from 75 to 95%, and nearly perfect performance (99.9%) when all sensors are used together. Unlike the decision tree, the drop in accuracy when one sensor is used is less pronounced. The ability of DCNN to process raw data directly allows it to learn richer and more relevant features for the task without requiring explicit feature extraction such as PCA.

The analysis of individual sensor performance reveals that Sensors 1 and 2 consistently outperform the others across the decision tree and DCNN models, respectively. For the decision tree model, which relies on features generated via PCA, Sensor 1 achieves the highest accuracy, followed by moderate performance from Sensors 2 and 3, while Sensors 6, 7, and 8 show significantly lower accuracy. The DCNN model, which operates on raw data, achieves superior performance with Sensor 2, with an accuracy of over 95%. Sensors 3 and 4 also demonstrate strong performance, while Sensors 6, 7, and 8 exhibit weaker results, indicating that they capture less relevant or redundant information for the detection task.

These findings suggest that while PCA and decision trees may offer insights with reduced computational overhead, the ability of the DCNN to process raw data makes it a more robust solution. Sensor 2 is the most critical for optimal event detection, and combining it with Sensors 3 and 4 could provide an effective trade-off between accuracy and sensor count.

Descriptions on the dataset indicate that beginning with position 3, the propagated gases are concentrated and well distributed [4]. This would explain that it is at these positions (positions ≥ 3) that we recorded the most relevant information for the identification of each gas. Hence, at these positions, the best accuracy was obtained using the data from each type of sensor.

Using multi-sensor data fusion to detect each gas resulted in more accurate models with consistent accuracy across all positions. This means that during the event detection process, people may not need to worry about the exact placement of the sensors if a wide range of sensors is available for data collection. Since each sensor provides different data, combining these pieces enables accurate event detection.

### 5.2. Results of the NSGA-II Multi-Objective Optimization

The NSGA-II algorithm was run on data derived from DCNN performance, specifically, the accuracy (Equation (Equation 5)) of the models was trained on individual sensor data to find compromised solutions between the number of sensors activated and overall detection accuracy. Equations (Equation 2) and (Equation 3) were optimized simultaneously.

Table 1 presents the results of NSGA-II, representing the solutions of the optimization process. Each solution specifies the recommended sensors at each position (P1–P5), identified to maximize detection accuracy while minimizing the total number of sensors deployed. Overall accuracy (Equation (Equation 3)) remains consistently high for all solutions, ranging from 94.46% to 94.61%, testifying to the robustness of the optimization process. These solutions contain both dominated and non-dominated solutions. After obtaining the output of NSGA-II, the dominated solutions are filtered using a Pareto dominance check to ensure the selection of the non-dominated solutions. The red highlights represent the Pareto-optimal or non-dominated solutions that form the Pareto front.

The table highlights a subset of sensors (from the total set of 8 evaluated sensors) best suited for gas detection at each position. The numbers listed under each position (e.g., “2, 4, 7”) indicate the sensors recommended at that position in the corresponding solution. The total number of sensors needed for all the positions in a given solution varies between 10 and 12 sensors across the configurations, with over 40 sensors considered for all positions combined.

Figure 8 shows all solutions represented by the blue points. The red points represent the Pareto front. It is the set of solutions that achieve the best trade-off between the number of sensors (Equation (Equation 2)) and overall accuracy (Equation (Equation 3)). The *X*-axis gives the total number of sensors used in the solution, and the Y-axis shows the overall accuracy percentage. The Pareto front shows that increasing the number of sensors generally improves precision but also highlights diminishing returns or points where additional sensors may not significantly enhance precision. This visualization helps identify optimal solutions based on deployment constraints (e.g., minimizing the number of sensors while maintaining acceptable precision).

### 5.3. Performance of the DCNN Using the Optimal Sets of Sensor

New detection models were trained with input data derived from the fusion of sensor combinations resulting from the optimization process (Section 4.5). Table 2 presents the accuracies obtained at different positions using models trained on sensor combinations recommended by the optimization process. The green cells indicate the highest accuracies achieved at each position. A key focus of this section is to evaluate how the selected sensors perform in ensuring high detection accuracy at each position. The following points can be highlighted.

The results confirm that the combination of multiple sensors can yield outstanding performance. In particular, at Position P3, the combination of Sensors (2, 3, 4) achieves the maximum accuracy of 100%, demonstrating the strength of this configuration for gas detection. At Position P2, the combination of Sensors (2, 4) achieves a very high accuracy of 99.9%, reinforcing the utility of using selected sensor subsets for specific positions.Interestingly, a single sensor (Sensor 2) alone achieves a remarkable 99.9% accuracy at Position P4, suggesting that in some cases, a single sensor can suffice without the need for fusion with other sensors. This reinforces the possibility of cost-effective configurations where single sensors perform optimally.The table highlights the variability of optimal sensor combinations across positions. For example, at Position P3, a multi-sensor configuration (2, 3, 4) leads to the best result. In contrast, at Position P4, a single sensor (Sensor 2) achieves nearly perfect accuracy. This underscores the importance of tailoring sensor combinations to the specific requirements of each position.Our previous analyses of DCNN performance (Section 5.1) noted that Sensor 2 was a critical contributor across different combinations, particularly when combined with Sensors 3 and 4, which may have a strong potential for optimal performance. The current results further support this observation by demonstrating that the (2, 3, 4) combination is indeed effective, achieving 100% accuracy at P3.

Table 2 validates and complements the insights from the DCNN analysis. It confirms that Sensor 2, especially in combination with Sensors 3 and 4, provides an excellent balance of accuracy and reliability across various positions. It also demonstrates the flexibility of configurations, where single sensors and multi-sensor setups can achieve near-optimal results depending on the positional requirements. These findings solidify the role of Sensor 2 as a key element in the optimization process, while highlighting the adaptability of sensor fusion for high-accuracy gas detection.

## 6. Conclusions and Future Work

This study demonstrates the significant impact that sensor location and quantity have on the accuracy of event detection in industrial environments. Using an online dataset of ten gases released at five distinct locations, we evaluated two supervised learning algorithms, DCNN and DT, trained both on single-sensor data and multi-sensor data fusion. While multi-sensor models achieved 100% accuracy, single-sensor models varied in accuracy from 80% to 97%, with higher accuracies observed at locations with greater gas concentration. These results confirm that sensor position has a direct influence on detection performance and that models achieving 100% accuracy generally require all sensors, that is, a total of eight for each of the nine measurement system modules, which can be costly in industrial applications.

To address this challenge, we implemented the NSGA-II multi-objective optimization algorithm to determine optimal sensor placement. We tried to minimize the number of sensors used while maximizing the detection accuracy. This approach offers a balanced solution by highlighting effective sensor combinations and positions that achieve high accuracy with fewer sensors. The results show that combining the NSGA-II algorithm with the performance of DCNN significantly optimizes sensor configuration. The number of sensors required can be reduced by 60–70% without a significant loss of accuracy. We have achieved 100% accuracy with only three sensors per module. This reduction underscores the potential savings and performance improvements that can be achieved through optimal sensor placement, even in complex detection scenarios.

The results of this study, particularly with respect to the impact of sensor placement and data fusion, can be extended to modern MOX sensors that incorporate multiple sensing elements in a single module. For example, the MiCS-6814 [42] module provides three separate gas sensors in a single package, which could be exploited using our optimization approach to select the most informative sensor combinations dynamically. Similarly, digital MOX sensors such as the Sensirion SGP41 [43] offer better drift compensation and long-term stability, addressing common problems with traditional MOX sensors. Although our current study focuses on an array of single-channel MOX sensors, future work could explore the effectiveness of our approach when applied to these next-generation sensing platforms. This could further improve the cost effectiveness and reliability of gas detection in industrial applications. Another promising direction for future research is the application of event detection and sensor optimization concepts to predictive maintenance and industrial machine diagnostics. Using optimized sensor placement strategies and detection algorithms, similar methods could be adapted to monitor critical machine parameters, such as vibration, temperature, or acoustic emissions, which are key indicators of equipment health. This approach would enable the precise tracking of machine wear and early fault detection, optimizing maintenance schedules and reducing downtime. The transposition of sensor optimization frameworks to predictive maintenance could provide a cost-effective solution to improve the reliability and performance of industrial assets.

## Figures and Tables

**Figure 1 sensors-25-02397-f001:**
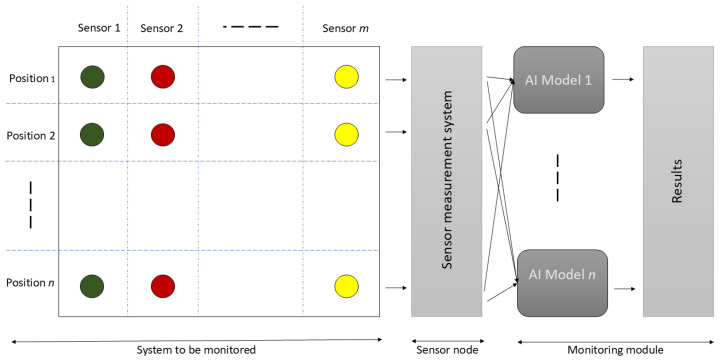
Scenario (AI is artificial intelligence; the circles represent sensors; the various colors show that the sensors differ).

**Figure 2 sensors-25-02397-f002:**
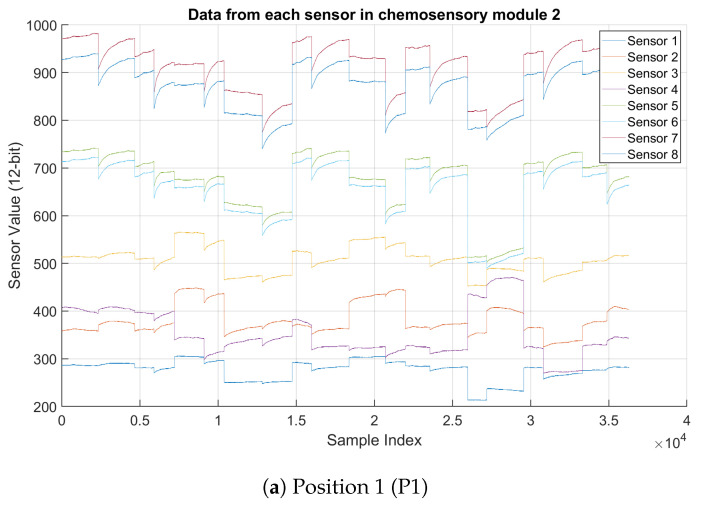
Data recorded from each sensor in chemosensory Module 2 at Positions 1 and 4, respectively.

**Figure 3 sensors-25-02397-f003:**
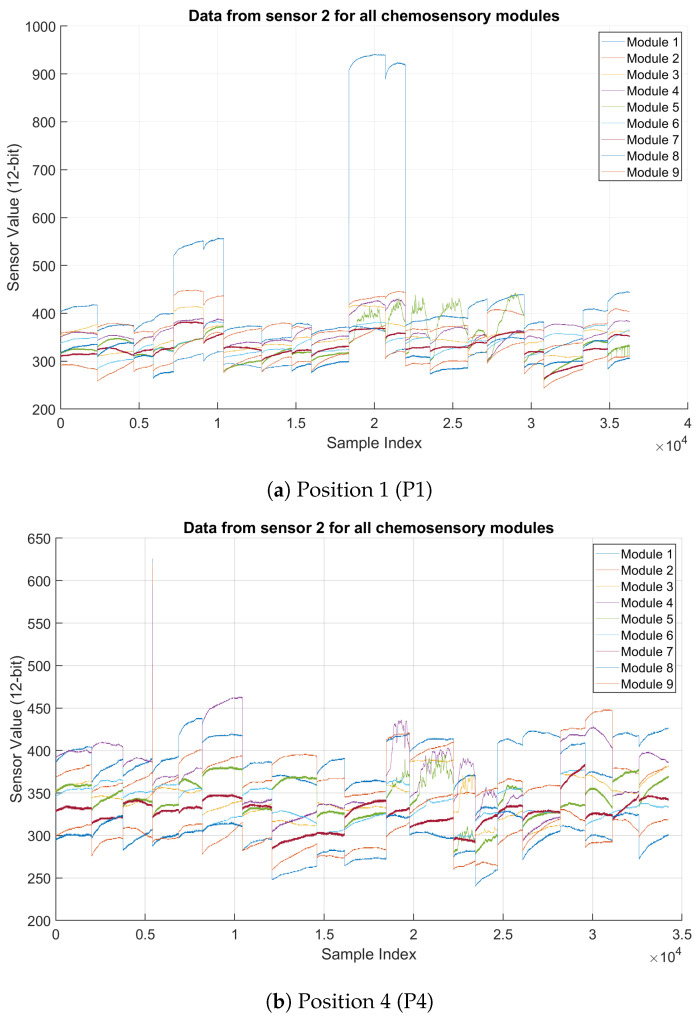
Data recorded from sensor 2 on each chemosensory module at Positions 1 and 4, respectively.

**Figure 4 sensors-25-02397-f004:**
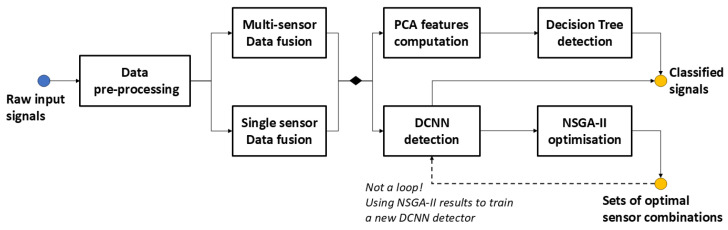
Experimental workflow for gas detection and sensor optimization. Raw signals undergo pre-processing, followed by either single-sensor fusion or multi-sensor fusion. Features are extracted using PCA for decision tree detection, while DCNN uses raw signals. NSGA-II optimization improves sensor utilization.

**Figure 5 sensors-25-02397-f005:**
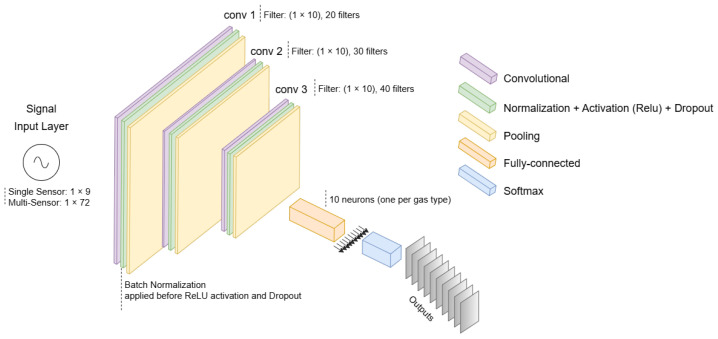
Deep convolutional neural network architecture.

**Figure 6 sensors-25-02397-f006:**
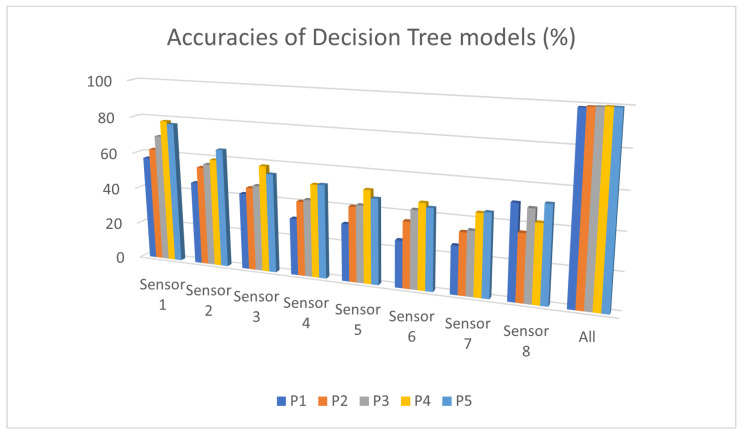
DT accuracies (P1–P5: position 1–position 5).

**Figure 7 sensors-25-02397-f007:**
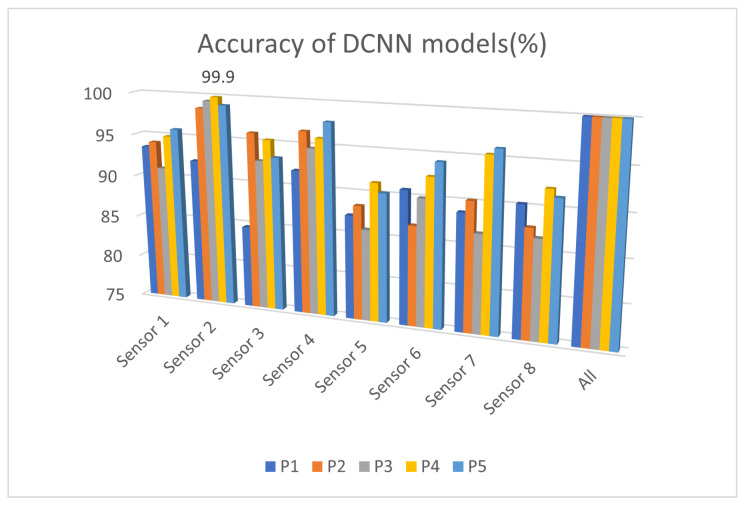
DCNN accuracies (P1–P5: position 1–position 5).

**Figure 8 sensors-25-02397-f008:**
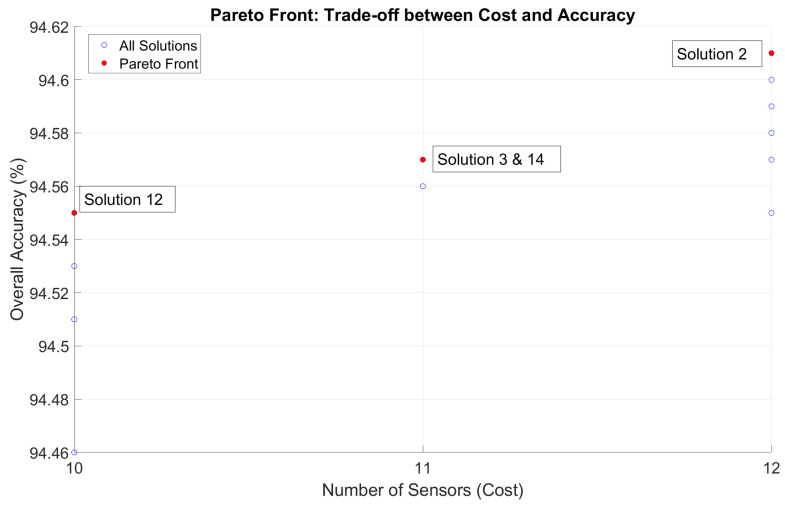
Pareto front: Pareto-optimal solutions resulting from the NSGA-II algorithm. They represent the best compromises between objectives and the true “non-dominated” set.

**Table 1 sensors-25-02397-t001:** Table of sensors selected by each solution at the various positions.

Positions (Pi)/ Solutions (S_t)	P1	P2	P3	P4	P5	Total Number of Sensors	Overall Accuracy (%)
**S_1**	2; 4	2; 4	2; 3; 4	2	2; 4; 7	11	94.56
**S_2**	1; 2; 4	2; 4	2; 3; 4	2	2; 4; 7	12	94.61
**S_3**	2; 4	2; 4	2; 3; 4	2	2; 4; 7	11	94.57
**S_4**	1; 2; 4	2; 4	2; 3; 4	2	2; 4; 7	12	94.60
**S_5**	1; 2; 4	2; 4	2; 3; 4	2	2; 4; 7	12	94.58
**S_6**	2; 4	2; 4	2; 3; 4	2	2; 4; 7	11	94.56
**S_7**	1; 2; 4	2; 4	2; 3; 4	2	2; 4; 7	12	94.60
**S_8**	2; 4	2	2; 3; 4	2	2; 4; 7	10	94.51
**S_9**	1; 2; 4	2; 4	2; 3; 4	2	2; 4; 7	12	94.60
**S_10**	2; 4	2	2; 3; 4	2	2; 4; 7	10	94.51
**S_11**	1; 2; 4	2; 4	2; 3; 4	2	2; 4; 7	12	94.57
**S_12**	2; 4	2; 4	2; 4	2	2; 4; 7	10	94.55
**S_13**	2; 4	2	2; 3; 4	2	2; 4; 7	10	94.53
**S_14**	1; 2; 4	2; 4	2; 4	2	2; 4; 7	11	94.57
**S_15**	2; 4	2	2; 3; 4	2	2; 4; 7	10	94.46
**S_16**	1; 2; 4	2; 4	2; 3; 4	2	2; 4; 7	12	94.55
**S_17**	1; 2; 4	2; 4	2; 3; 4	2	2; 4; 7	12	94.59

**Table 2 sensors-25-02397-t002:** Accuracy (Equation (Equation 5)) at each position of models trained using data from the fusion of sensors selected during the optimization process.

Sensors/Positions (Pi)	Sensor 2	Sensors (2; 4)	Sensors (2; 3; 4)	Sensors (2; 4; 7)	Sensors (1; 2; 4)
P1	-	98.6%	-	-	97.8%
P2	97.8%	99.9%	-	-	-
P3	-	99.7%	100%	-	-
P4	99.9%	-	-	-	-
P5	-	-	-	99.1%	-

## Data Availability

Data are contained within the article.

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
