# Peer review of "Optimizing Sensor Placement for Event Detection: A Case Study in Gaseous Chemical Detection"

_sensors, 2025, doi:10.3390/s25082397_

Round 1
Reviewer 1 Report
Comments and Suggestions for Authors
This work is focused not on sensor itself but on the deployment of sensors. They adopted an algorithm as called NSGA-II to reduced the number of sensors while maintain the detection accuracy. Generally, the method reported here might be valuable for some IOT scenarios where large number of sensors is required. But the results obtained in this work is based on a specific open-sourced gas-sensor dataset, which is mostly designed for the gas sensing community for classification method development. The results reference value is quite limited. Here is some detailed comments :
- A diagram depicting the topological structure of the DCNN is highly recommended.
- How the accuracy of DCNN corresponding to a single sensor be calculated, since there are 72 inputs in the input layer of the DCNN? By silence all the other 71 sensors? The authors should provide more information about their methodology!
- NSGA-II should be described in more details rather than simply give a reference since it is an essential algorithm.
Reviewer 2 Report
Comments and Suggestions for Authors
Dear Authors
Thank you for manuscript, I had read with interests. I won't criticize the work "On the performance of gas sensor arrays in open sampling systems using Inhibitory Support Vector Machines" now. This work was done more than 10 years ago, at the current level of technology (There are now on open market more stable MOX sensors with digital output.). I have a few technical comments to your manuscript:
- Please, check all links in manuscript text. I did not find link to article [35]. The reference numbers must go in strict sequence one after another with a cumulative total. This is not the case with your manuscript (please, look how starting introduction chapter)!
- The data on which your experiment is based is relatively old - 2013. Not all readers have access to the old original article text. In your case, I would devote the introductory chapter to a brief description of the experiment and the Figaro sensors used in one. This will give the reader a better understanding of the algorithms you used. Because now, to understand, readers need to have two articles in front of their eyes - one original experimental, the second yours. This is difficult to do fast in parallel!
- It can be added to the conclusions that there are already systems with three MOX sensors in one package, for example - https://www.sgxsensortech.com/sensor/mics-6814 which can use your developed algorithm on these sensors with a certain percentage of success (under discussion). Readers can use your approach and these sensors to build their own systems, as opposed to, for example, digital sensors with a closed algorithm - https://sensirion.com/products/catalog/SGP41 (This design represents at least four different MOX sensors inside for increasing stability and sensitivity. Please looks https://www.mdpi.com/1424-8220/18/4/1052 )
Round 2
Reviewer 1 Report
Comments and Suggestions for Authors
The authors have revised the manuscript properly. Now, I would recommend a publication.